# Chemical Composition Analysis, Cytotoxic, Antimicrobial and Antioxidant Activities of *Physalis angulata* L.: A Comparative Study of Leaves and Fruit

**DOI:** 10.3390/molecules27051480

**Published:** 2022-02-22

**Authors:** Jayachithra Ramakrishna Pillai, Adil Farooq Wali, Godfred Antony Menezes, Muneeb U. Rehman, Tanveer A. Wani, Azher Arafah, Seema Zargar, Tahir Maqbool Mir

**Affiliations:** 1Department of Pharmaceutical Chemistry, RAK College of Pharmaceutical Sciences, RAK Medical and Health Sciences University, Ras Al Khaimah 11172, United Arab Emirates; jayachithra@rakmhsu.ac.ae; 2Department of Microbiology, RAKCOMS, RAK Medical and Health Sciences University, Ras Al Khaimah 11172, United Arab Emirates; godfred@rakmhsu.ac.ae; 3Department of Clinical Pharmacy, College of Pharmacy, King Saud University, Riyadh 11451, Saudi Arabia; muneebjh@gmail.com (M.U.R.); aazher@ksu.edu.sa (A.A.); 4Department of Pharmaceutical Chemistry, College of Pharmacy, King Saud University, Riyadh 11451, Saudi Arabia; twani@ksu.edu.sa; 5Department of Biochemistry, College of Science, King Saud University, Riyadh 11451, Saudi Arabia; szargar@ksu.edu.sa; 6National Center for Natural Products Research, Research Institute of Pharmaceutical Sciences, School of Pharmacy, University of Mississippi, Mississippi, MS 38677, USA; tmmir@olemiss.edu

**Keywords:** *Physalis angulata*, Solanaceae, antioxidant activity, cytotoxic activity, HeLa, MCF-7, DLD-1

## Abstract

*Physalis angulata* L. belongs to the family Solanaceae and is distributed throughout the tropical and subtropical regions. *Physalis angulata* leaf and fruit extracts were assessed for in vitro anticancer, antioxidant activity, and total phenolic and flavonoid content. The GC-MS technique investigated the chemical composition and structure of bioactive chemicals reported in extracts. The anticancer activity results revealed a decrease in the percentage of anticancer cells’ viability in a concentration- and time-dependent way. We also noticed morphological alterations in the cells, which we believe are related to *Physalis angulata* extracts. Under light microscopy, we observed that as the concentration of ethanolic extract (fruit and leaves) treated HeLa cells increased, the number of cells began to decrease.

## 1. Introduction

Plants with medicinal properties serve as the basic treatment for many ailments in cultural diversities worldwide. These medicinal plants exhibit various activities against many major diseases and their active constituents. In recent years, phytoscience has made significant contributions to modern medicine. As a result, medicinal plants are the primary source of treatment for a wide range of health issues. Another reason is that herbal treatments are readily available and accessible and are very inexpensive. In several nations, people in rural areas still rely on herbal medicines to treat all of their health problems. Phytomedicine is widely used in modern medicine to treat various disorders, including diabetes and cancer. Many studies have been published on the use of phytomedicine in cancer treatment. According to the World Health Organization (WHO), herbal medicine is used by 60% of the world’s population, and about 80% of the people in emerging nations rely on it nearly entirely for their basic health care requirements. [1,2,3].

*Physalis angulata* (Figure 1) belongs to the family Solanaceae and includes more than 120 species with different herbal characteristics and perennial habits [4]. This plant is a bushy annual herb that grows to about 50 cm in height, and is glabrous or with minute simple hairs [5]. It has bell-shaped flowers and balloon-like fruit which drops downwards. The fruit of *Physalis angulata* are edible and very tasty [6]. This herb is distributed widely throughout the tropical and subtropical regions. It is known by different names such as camapu; cut leaf groundcherry; wild tomato, winter cherry, etc. A literature survey revealed that *Physalis angulata* could be used as an herbal medicine for many ailments. It is widely known to be a stimulant for the immune system. It can be used in food, mainly in preparing sauces [7,8]. In several regions of the World, the extracts or infusions of this plant are used as anti-malarial, anti-asthmatic, and for dermatitis. The isolated phytoconstituents of *Physalis angulata* exhibited antitumor activities against many cancer cell lines [9,10] by in vitro studies. It has been traditionally used for antipyretic purposes in Japan. This plant has been used in intestinal and digestive problems and for many other ailments like sores, boils, cuts, etc. The leaves of the plant are used in salads. Studies proved that *Physalis angulata* exhibits many therapeutic activities like antiallergic, antiasthmatic, antileishmanial, antmalarial, and immunomodulatory activity [11,12,13]. The phytochemical investigation of *Physalis angulata* showed many primary and secondary metabolites such as carbohydrates, minerals, vitamins, lipids, and phytosterols. The entire plant contains various steroidal lactones which belong to physaline, and withanolide such as physalins A–I, physagulin A–G, withangulatin A, and withanolide T. Withanolides have a C-28 ergostane-type steroid structure with a δ-lactone group at C-22 and C-26. It also contains flavonol glycoside named myricetin 3-O-neohesperidoside [14,15,16,17,18].

Cancer is the term used for diseases when abnormal cells multiply without any control in any body organ or tissue and attack the nearby body parts. If the spread is uncontrollable, it can cause death. It is considered the second foremost reason for death around the world. The common types of cancer found mostly in men are lung, liver, colorectal, prostate, and stomach. In women, breast, lung, colorectal, cervical, and thyroid cancers are mostly found [19].

Much recent research has focused on plant-derived chemoprotective constituents. These active constituents can act as target specific for malignant cells or can prevent cancerous cell multiplication [20]. The current cancer treatment is associated with severe side effects, including the damage of normal body cells and many others. Therefore, the need for safe drugs or methods for cancer treatment is in high demand. The current study on the leaves and fruit of *Physalis angulata* includes the cytotoxic activity on the breast cancer, colorectal and cervical cancer cell lines.

## 2. Results

### 2.1. Preliminary Phytochemical Screening

The preliminary phytochemical screening of the *Physalis angulata* fruit and leaves proved that it contains various secondary metabolites such as alkaloids, glycosides, flavonoids, tannins, and phenolics.

### 2.2. Anti-Oxidant Activities

The DPPH and hydrogen peroxide assays of all the extracts of the fruit and leaves of *Physalis angulata* exhibited remarkable radical scavenging activity. The standard used to compare the results was ascorbic acid.

Even though all the extracts showed activity, the ethanolic extracts of *Physalis angulata* leaves and fruit had very good activity for both the assays. The ethanolic extract of the leaves showed 85% and 81% radical scavenging activity upon a standard value of ascorbic acid with 98% in DPPH and hydrogen peroxide assays, respectively, as illustrated in Figure 2A,B.

Similarly, the ethanolic extracts of the fruit exhibited 86% and 80.9% radical scavenging activity upon a standard value of ascorbic acid with 98% in DPPH and hydrogen peroxide assays, respectively, which is illustrated in Figure 2B,D.

### 2.3. Total Phenolic Content (TPC) and Total Flavonoid Content (TFC)

The TPC and TFC of all the leaves and fruit extracts were evaluated. The TPC of all the extracts of the leaves were 54.4 ± 3.4, 96.7 ± 4.5, and 140.65 ± 3.8 GAE/g respectively, with a calibration curve of R2 0.998, whereas the TFC with R2 0.998 was 210.32± 3.6, 238.24 ± 4.4, and 370.64 ± 4.33 QE/g for petroleum ether, ethyl acetate, and ethanolic extracts, respectively (Table 1).

The TPC of all the three extracts of the fruit were 75.3 ± 4.6, 69.4 ± 3.6, and 106.5 ± 3.5 GAE/g, respectively, with a calibration curve of R2 0.998, whereas the TFC with R2 0.998 was 40.78 ± 4.3, 100.83 ± 4.2, and 130.4 ± 2.6 QE/g for petroleum ether, ethyl acetate, and ethanolic extracts, respectively (Table 2).

### 2.4. Chemical Profiling of Ethanolic Extracts (Fruit and Leaves) of Physalis Angulata by Using GC-MS Analysis

The results of a GC-MS analysis of *Physalis angulata* L. ethanolic extracts (fruit and leaves) led to the identification of several chemicals. Table 3 (ethanolic extract fruit) and Table 4 (ethanolic extract leaves) compares the retention time, molecular formula, and molecular mass (m/z) of chemical compounds with various databases, literature surveys, and research publications. The chemical components found in the ethanolic extracts (fruit and leaves) of *Physalis angulata* L. presented in Table 3 and Table 4 are based on the identification data. Appendix A show a representative full scan chromatographic profile of *Physalis angulata* ethanolic extract (leaves and fruits) (Appendix A).

### 2.5. Elemental Analysis

The dried leaf and fruit samples of *Physalis angulata* were analyzed by ICP OES. The relative standard deviation for the different samples was determined for the precision of the procedure. *Physalis angulata* leaf and fruit contains various macro and micronutrients. The leaf is rich in calcium and potassium when compared to fruit. The leaf contains 24,503 mg/Kg calcium, iron 3338.8 mg/Kg, and 35,725 mg/Kg potassium. In comparison, the fruit contains 4143 mg/Kg calcium, 478.9 mg/Kg iron, and 32,856 mg/Kg potassium (Table 5).

### 2.6. Cytotoxic Studies

The cytotoxic activity of ethanolic extracts of leaves and fruit of *Physalis angulata* was determined by the MTT cell viability assay using HeLa, DLD-1, and MCF-7 cell lines. The cell viability was decreased with both extracts depending on the dose. The MTT cell viability assay was carried out in different fruit and leaf extracts (6.25, 12.5, 25. 50, and 100 μg/mL) for all three cell lines. The calculated cell viability was decreased with an increase in extract concentration. The percentage viability of *Physalis angulata* leaf extracts at 100 μg/mL was observed at 46.23, 33.66, and 51.54 for DLD-1, HeLa, and MCF-7 cell lines, respectively (Figure 3). The percentage viability of *Physalis angulata* fruit extracts at 100 μg/mL was observed 70, 69.41, and 65.27 for DLD-1, HeLa, and MCF-7 cell lines, respectively. The fruit extracts’ LC50 value (calculated using ED50 PLUS V1.0 software) were 188, 167, and 157 μg/mL for DLD-1, HeLa, and MCF-7 cell lines, respectively. The same for the leaf extracts of *Physalis angulata* were 90, 44, and 100 μg/mL for DLD-1, Hela, and MCF-7 cell lines, respectively. The MTT assay of the ethanolic extracts of *Physalis angulata* fruit and leaves revealed that the leaf extracts have significant cytotoxic activity on different cell lines.

### 2.7. Antibacterial Potential

Using the serial dilution technique, *Physalis angulata* leaves and fruit extracts were evaluated for their antibacterial activity. The results show that *Physalis angulata* leaves and fruit extracts possess moderate activity on these pathogens. Among the MIC/MBC of the leaf extracts against *E. coli* (ATCC 25922), the ethyl acetoacetate extracts possess moderate activity with less concentration. Similar results were observed for fruit extracts also. Among the MIC/MBC of the leaf and fruit extracts against *S. aureus* (ATCC 25923), the ethyl acetoacetate extracts also showed moderate activity against this pathogen (Table 6). Appendix A represents the MIC values plates (Appendix A). All the results were compared against the standard gentamycin.

## 3. Discussion

The application of medicinal plants for managing various ailments is worldwide now. It varies from traditional to popular medicines of each country using standardized medicinal plant extracts. Natural medicines possess a very high therapeutic index against different tumor cells, and they have been proved to control the abnormal multiplication of cells [21,22]. This study gives information about the phytoconstituents, TFC, and TPC content of *Physalis angulata* leaf and fruit extracts with the in-vitro antioxidant, antimicrobial and cytotoxic activities.

Free radicals are the major reason for many malignancies, diabetes, degenerative diseases, etc. [23]. These free radicals are produced due to the lack of a natural antioxidant defense mechanism [24]. Many phytonutrients from plants have shown radical scavenging activity against these free radicals [25]. These phytonutrients are in high demand now [24]; the phytonutrient flavonoids are polyphenols. Almost all types of plants contain flavonoids [23]. Recent studies show that flavonoids and phenolics are powerful antioxidants that improve the immune system. Diet, rich in flavonoids, can prevent cancer, neurodegenerative, and cardiovascular diseases [18,26]. Many studies show that women with higher levels of flavonoid intake through diet will have a lower risk of developing breast cancer [27]. Many studies have proved that the flavonoids and phenolic content in the plant material play an important role against oxidative stress [26]. Scavenging activities are facilitated by the presence of the number and position of hydroxyl and methoxy groups in the phenolic rings [28].

DPPH and hydrogen peroxide assays were used in this study to assess the antioxidant activities of the various extracts. These tests are very common and accurate to determine the radical scavenging activity due to the hydrogen donating capacity to the free radicals. The ethanolic extracts of *Physalis angulata* leaves and fruit showed significant antioxidant activity. The total phenolic content and flavonoid content of the leaf extracts of *Physalis angulata* was higher than that of the fruit extracts. In the leaf extracts, ethanolic extracts showed the maximum TPC and TFC. The significant antioxidant activity shown by the leaf extracts may be because of the higher levels of phenolics and flavonoids present in them [29,30].

Bioactive components of medicinal plants have been used to manage cancer for many years [31]. The cytotoxic activity of the herbal drugs against various types of cancer has been established by several clinical studies and phytochemical screening [32,33]. The main concern with modern cancer chemotherapy is the drug resistance that develops during the treatment [34]. Many studies have shown that natural compounds separated or derived from plants have a significant potential for cancer prevention [27]. The cytotoxic activities of polyphenols have been reported in various studies against different cancer cells [35]. The polyphenols are capable of initiating apoptosis, which is the reason for their anticancer activities [36].

Since the TPC and TFC were relatively high in the ethanolic extracts, and due to this extract’s significantly high antioxidant activity, we chose the ethanolic extract for the cytotoxic studies. The ethanolic extracts of *Physalis angulata* fruit and leaves were evaluated against three different cell lines such as HeLa, DLD-1, and MCF-7. The MTT assay method was used to detect the cytotoxic activity of the plant extracts. The percentage cell viability decreased significantly with increased sample concentration with the ethanolic extracts of fruit and leaves of *Physalis angulata*. When comparing the results of fruit and leaves, the ethanolic extracts of the leaves of Physalis angulata showed more significant cytotoxic activity than fruit extracts. When tested against HeLa, DLD-1 and MCF-7 cell lines, the ethanolic extracts of the leaf showed excellent activity with HeLa with a low LC_50_ (44 ± 10µg/mL) value.

Our study shows that the leaves and fruit of *Physalis angulata* are rich with micro and macro elements, which are highly important for the body as medicine and food supplements. Many studies have shown that these nutrients can help the body balance blood pressure, normalize increased cholesterol levels, protect the liver, promote immunity, etc. [37,38]. Along with the phytoconstituents present in the leaves and the fruit, this also might be the reason for the therapeutic activities of *Physalis angulata*.

The estimation of *Physalis angulata* leaves’ antibacterial activity and fruit extracts demonstrated reasonable activity against both Gram-negative and Gram-positive bacteria [39]. Many studies showed that the polyphenols present in the plant possess antibacterial activity. *Physalis angulata* plant also contains a good amount of polyphenols [40]. These reports correlate with the current findings. The polyphenols present in *Physalis angulata* can cause its antibacterial activity [41].

Various chemical compounds were identified using GC-MS in the methanolic extract of Physalis angulata leaves and fruit. The major chemical compounds were identified from the fruit extract linoleic acid, such as Cho-lest-5-ene-16,22-dione, 3β,26-dihydroxy-, 3-acetate, (20S,25R)-, Lup-20(29)-en-3-ol, acetate, (3β)- and α-Tocopherol, whereas Hexahydrofarnesyl acetone, 3,7,11,15-Tetramethyl-2-hexadecen-1-ol, Phytol, Oleic Acid, Octadecanoic acid, 9,12-Octadecadienoic acid (Z,Z)-, 9-Octadecenoic acid (Z)-, 2-hydroxy-1-(hydroxymethyl) ethyl ester, and Ergost-5-en-3-ol, acetate, (3β,24R) were identified from the leaves extract. In this study, the higher antioxidant and antimicrobial activities of the methanol extract could be correlated to a higher number of bioactive compounds, including Lauric acid, Oleic acid, Hexahydrofarnesyl acetone, Oleic acid, n-hexadecanoic acid, Phytol, and α-Tocopherol. Similarly, the occurrence of these compounds is also evidenced in many medicinal plant leaves extracted with methanol [42]. Lauric acid is a saturated fatty acid reported to have antimicrobial activity against various bacterial strains [42]. Linoleic acid is polyunsaturated fatty acid reported in many studies to inhibit the growth of all the strains of Gram-positive bacteria [43]. One of the most naturally occurring biologically active forms of Vitamin E is alpha-tocopherol. It’s a known antioxidant. The antibacterial and cytotoxic activities of α- tocopherol are reported in research studies [44,45]. Hexadecanoic acid and its derivatives exhibit strong antimicrobial and anti-inflammatory activity [46,47]. Phytol is a natural antioxidant and possesses various pharmacological importance. Phytol is an important diterpene that possesses antimicrobial, antioxidant, and anticancer activities [48]. Oleic acid is an omega-9 fatty acid produced in the body. It has been reported that the oleic acid present in various plants can exhibit apoptotic and antibacterial activity [49,50]. The compound stigmasterol in hexane extract is also reported to have various biological activities [51].

## 4. Materials and Methods

### 4.1. Plant Collection and Preparation of the Plant Extract

The *Physalis angulata* L. (Syn.: P. minima L.; P. longifolia) plant was collected from Kerala, India. The plant was then authenticated by Dr. T. Sabu, JNTBGRI, Thiruvananthapuram, India. A voucher herbarium specimen (No. 852016) was deposited in JNTBGRI herbarium (TBGT). The leaf and fruit of *Physalis angulata* were used for the current study. The leaves and fruit were separated from the plant, shade dried, and powdered. 60 g each of leaf and fruit powder were taken in two different containers, and 500 mL of petroleum ether was added to each. The extraction was done by the ultra-sonication method. After sonication for 1 h, the extract was collected and filtered. The leaf and fruit powders were dried and extracted again with 500 mL of ethyl acetate under sonication. After extraction with ethyl acetate, it was filtered again, and the extracts were collected. The dried plant parts were extracted again with ethanol and collected the extract. After the solvent evaporation, the leaf extracts weighed around 5 g, and the fruit extracts weighed around 4 g. These extracts were used for various activities and phytochemical screening. The flowchart of Section 4 is represented by Appendix A).

### 4.2. Preliminary Phytochemical Evaluation

The extracts of *Physalis angulata* fruit and leaves were subjected to preliminary phytochemical screening using the standard protocol [52]. It was found that various major secondary metabolites are present in the leaf and fruit.

### 4.3. In Vitro Anti-Oxidant Screening

The in vitro antioxidant screening of the leaf and fruit extracts of *Physalis angulata* was done by spectrophotometric methods such as DPPH and Hydrogen peroxide assays. The spectrophotometric assays depend on the reaction between the free radical and that of the antioxidant moiety that can donate a hydrogen atom. Ascorbic acid was used as the standard.

#### 4.3.1. DPPH Assay

The DPPH assay for the radical scavenging activity of both the leaves and the fruit extracts of Physalis angulata was done using the method described by Adil Wali et al. [53]. When the DPPH radical reacts with the hydrogen donor from the extracts, its color disappears. The absorbance was measured spectrophotometrically at 517 nm (Shimadzu, UV-1800). For each sample, the readings were repeated three times. A comparison of all these results was done with that of standard.

#### 4.3.2. Hydrogen Peroxide Assay

The hydrogen peroxide assay for the radical scavenging ability of both the leaves and the fruit extracts of *Physalis angulata* was done based on the method of Marchelak et al. [54]. The absorbance was measured spectrophotometrically at 230 nm (Shimadzu, UV-1800), and the blank solution was prepared with phosphate buffer without hydrogen peroxide. The readings were taken three times for each sample. A comparison of all these results was done with that of standard.

### 4.4. Determination of TPC

The leaves and the fruit extracts of Physalis angulata were tested for TPC (Total Phenolic Content) by the Folin- Ciocalteau method [55]. The calibration curve was plotted by mixing 1 mL of 0.5, 0.75, 1.0, 1.25, and 1.5 mg/mL Gallic acid solution with 2.5 mL of Folin- Ciocalteau reagent and 2.5 mL of sodium carbonate solution. The solutions were mixed thoroughly and kept for 30 min. The absorbance was measured at 725 nm using UV-Spectrophotometer (Shimadzu, UV-1800). 0.5 mL of all the extracts of both leaf and fruit of Physalis angulata (5 mg/mL) was mixed with the same reagents. After keeping for 1 h, the absorbance was measured. The TPC is calculated as follows:

Total phenolic content = cV/m

The total phenolic content is expressed as Gallic Acid Equivalent or GAE (mg/g),

c = gallic acid concentration calculated from the calibration curve

V = extract volume

m = mass of the plant extract (g).

Three readings were taken for each sample.

### 4.5. Determination of TFC

The TFC (the Total Flavonoid Content) of the leaves and the fruit extracts of *Physalis angulata* was determined spectrophotometrically by following the aluminum chloride method [56]. The calibration curve was generated by using the standard quercetin. The standard was prepared by mixing 1 mL each of 0.5, 0.75, 1.0, 1.25, and 1.5 mg/mL quercetin solution with ethanol and 10% aluminum chloride solution. This was mixed with 1 M potassium acetate and distilled water. The absorbance of these samples was taken after incubation for 3 min using a Shimadzu UV spectrophotometerUV-1800 at 415 nm. All the extracts (0.5 mL) of both the leaf and fruit of Physalis angulata were taken, and absorbance was measured. The TFC is calculated as follows:

Total flavonoid content (TFC) = cV/m

The total phenolic content is expressed as Quercetin equivalent (QE),

c = Quercetin concentration calculated from the calibration curve

V = extract volume

m = mass of the plant extract (g).

All the readings were taken in triplicates

### 4.6. Chemical Profiling of Ethanolic Extracts (Fruit and Leaves) of Physalis Angulata by Using GC-MS Analysis

The phytochemical investigation of ethanoic extracts (fruit and leaves) were performed on GC-MS equipment and tgas chromatography (Agilent7890 A GC) was done with a DB 5MS 30 m × 0.250 mm Diameter × 0.25 Micro Meter Thickness) interfaced with a 5675C Inert MSD with a Triple-Axis detector used for the GC-MS analysis. The carrier gas was helium, set to a column velocity flow of 1.0 mL/min.

Other GC-MS conditions include a 150 °C ion source, a 280 °C interface, a 7.0699 psi pressure, a 1.8 mm out time, and a 3 μL injector in split mode with a split ratio 1:50 and a 280 °C injection temperature. The column temperature was set to 50 °C for 5 min before gradually increasing to 80 e V at a rate of 4 °C/min. The temperature was raised to 280 °C at a 20 °C/min rate for 5 min. It took 75 min to complete the elution. All information, acquisitions, and evaluations were regulated using the National Institute of Standards and Technology (NIST 08) Spectral Database Library to validate the chemical compounds.

### 4.7. Elemental Analysis by ICP-OES

The elemental analysis of the dried leaf and fruit of Physalis angulata was done by a Spectrogenesis ICP-OES (Inductively coupled Plasma-Optical Emission spectrometer) After weighing, around 3 g of the dried fruit and leaf powder of Physalis angulata were taken for the analysis. These samples were digested using concentrated nitric acid (3 mL). It was then heated on a hot plate to digest the organic matter. After cooling, the samples were treated with concentrated hydrochloric acid (1 mL) and all of the samples were heated again and then and then cooled to room temperature. After filtration, these samples were transferred to a volumetric flask and, with distilled water, made up to the volume. This sample is used as the stock to evaluate the micro and macronutrients by ICP-OES. Proper dilutions were made, and a calibration curve was plotted accordingly. By free aspiration, the diluted samples were introduced into the ICP-OES. By associating with that of the prepared standard, the intensities of the micro and macro elements present in the test samples were calculated.

### 4.8. Evaluation of In-Vitro Cytotoxic Activity

The cytotoxic evaluation of the ethanolic extracts of the fruit and leaves of *Physalis angulata* was done by MTT assay. HeLa (Cervical carcinoma), DLD-1 (Human Colorectal Adeno carcinoma) and MCF-7 (Human Breast Adeno carcinoma) cells were procured from National Centre for Cell Sciences (NCCS), Pune, India and maintained Dulbecco’s modified Eagles medium, DMEM (Sigma Aldrich, St. Louis, MO, USA). This method uses a colorimetric method to assess the metabolic activity of cells [57]. This allows us to determine the number of live cells, their growth, and cytotoxicity. This method is based on metabolically active cells reducing tetrazolium salt, yellow color, to purple color crystals. The ethanolic extract of the leaf and fruit of *Physalis angulata* and the HeLa cells were tested on MCF-7, DLD-1. All the cell lines were cultured with DMEM, L-glutamine, 10% FBS, sodium bicarbonate, and an antibiotic solution with penicillin, streptomycin, and amphotericin B with the concentration of 100 U/mL, 100 µg/mL, and 2.5 µg/mL, respectively. All these cell lines were kept in a humidified incubator at 37 °C. An inverted phase-contrast microscope was used to evaluate the viability of cells and the MTT method of assay. The compound stock solution was prepared by mixing 1 mg of sample in 1 mL DMEM.

#### Cytotoxic Estimation

The growth medium was removed after 24 h. Each compound was freshly prepared and diluted five times. 100 µL concentrations were added three times to the respective wells and incubated and humidified. Control cells were also maintained untreated.

### 4.9. Estimation of Anti-Bacterial Activity

According to the standard procedure, *Physalis angulata* leaves and fruit extracts were evaluated for their antibacterial activity [58]. Inoculum containing 106 CFU/mL of Gram-negative and Gram-positive bacterial culture such as Escherichia coli and Staphylococcus aureus (ATCC 25922 and ATCC 25923, respectively) were used for the evaluation. All these strains were isolated from Ras Al Khaimah Medical and Health Sciences University’s clinical microbiology and immunology division. The diffusion method was used for the estimation. Wells (8 mm diameter) were holed into the Muller Hinton agar medium. These wells are then packed with *Physalis angulata* leaves and fruit extract (100 µL–25 mg/mL). Plates were diffused for 2 h at room temperature. A 1% DMSO (5 mg/500 µL) solution was used to prepare the stock solution of all the extracts of *Physalis angulata*. All their plates were kept vertically and incubated for twenty-four hours at 37 °C. Ethanol and gentamycin (10 μg) were used as negative control and standard/positive control, respectively. The growth inhibition zones of each extract were calculated after the incubation period. The test was repeated against both pathogens. Mean± standard deviation was used to represent the data.

The broth microdilution method (BMM) was used to determine the MIC (minimum inhibitory concentration) and MBC (minimum bactericidal concentration). BMM is a method used to quantify the in-vitro activity of an antibacterial agent. For BMM, the requirements are a sterile tray with various concentrations of antimicrobial agents of interest tested against a standardized number of microbial cells [59]. MICs were determined by observing the lowest concentration of an antimicrobial agent that would inhibit the visible growth of a microorganism after overnight incubation. The lowest concentration of an antimicrobial that prevents the growth of an organism after subculture onto antibiotic-free media gives the value for MBC [50].

## 5. Conclusions

The present study discussed the cytotoxic activity of the ethanolic extracts of *Physalis angulata* leaves and fruit. The various extracts of the leaf and fruit of *Physalis angulata* showed significant antioxidant activity. The cytotoxic activity was evaluated by the MTT assay method against various cell lines such as HeLa (cervical), DLD-1 (colon), and MCF-7 (breast). When comparing the results of the antioxidant activity of various extracts of *Physalis angulata* leaves and fruit, the ethanolic extract showed better activity than that of ascorbic acid. The TPC and TFC values were also high in ethanolic extracts of *Physalis angulata* compared with other extracts. When comparing the results of fruit and leaf extracts, the leaf extracts of *Physalis angulata* showed better antioxidant activity and TFC and TPC. The cytotoxic activity shown by the ethanolic extracts of *Physalis angulata* leaf was significantly better than that of fruit extracts of the plant. The ethanolic extract of the leaf exhibited very good anticancer activity against human HeLa cell lines through its low IC_50_ (44 ± 10 µg/mL) values. Thus, further investigations and isolation can be done for the leaf of *Physalis angulata*.

## Figures and Tables

**Figure 1 molecules-27-01480-f001:**
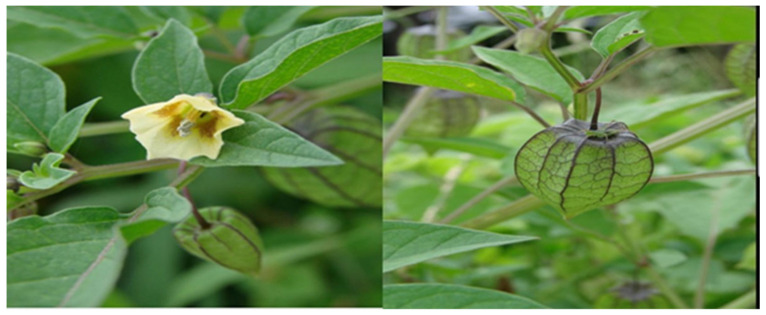
*Physalis angulata* (fruit and leaves) belongs to the family Solanaceae, includes more than 120 species.

**Figure 2 molecules-27-01480-f002:**
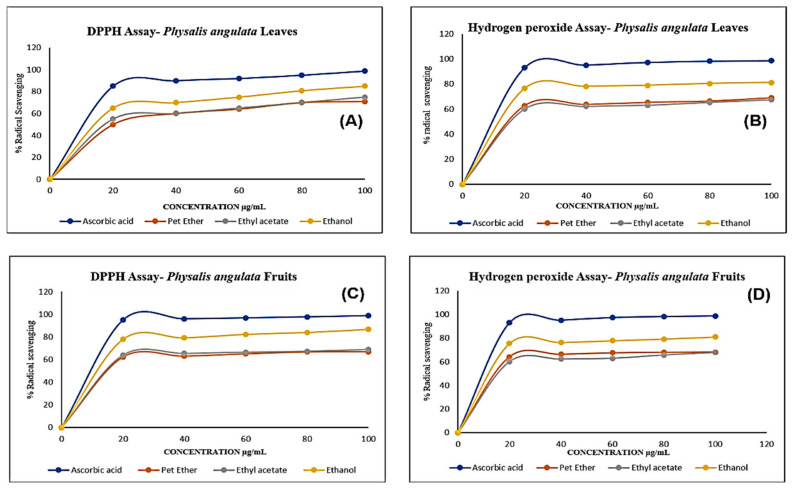
The antioxidant activity of Physalis angulata leaves and fruit extracts (**A**–**D**). The dose-dependent DPPH and H_2_O_2_ scavenging activity of leaves and fruit extracts relative to that of L-ascorbic acid. All results are presented as the mean  ± SD (*n*  =  3).

**Figure 3 molecules-27-01480-f003:**
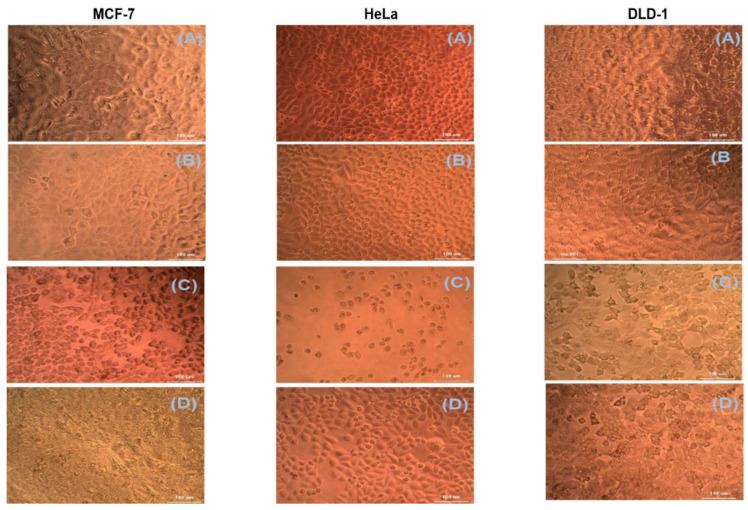
Effect of ethanolic extract of *Physalis angulata* leaves and fruit on different cancer cell lines. Impact of *Physalis angulata* extracts on morphological changes and viability of human cancer cells MCF-7, HeLa, and DLD-1. After 72 h of incubation, the photomicrograph shows the effect of different concentrations of extract of *Physalis angulata* leaf and fruit on MCF-7, HeLa, and DLD-1 cells. Photomicrograph (**A**): Blank control; (**B**): Positive control; (**C**): Low concentration; (**D**): High concentration. The data display the mean of the test as measured in triplicate. One-way ANOVA analysis to compare between control, 6.25 and 100 µg/mL of the extract.

**Table 1 molecules-27-01480-t001:** Total phenolics and flavonoids content of *Physalis angulata* Leaves.

Extract	TPC (* mg GAE/g)	TFC (** mg QE/g)
Petroleum ether	54.4 ± 3.4	210.32 ±3.6
Ethyl acetate	96.7 ± 4.5	238.24 ± 4.3
Ethanol	140.65 ± 3.8	370.64 ± 4.33

* Total phenolics content is expressed in terms of Gallic acid equivalent (μg of GA/g). ** Total flavonoids content is expressed in terms of quercetin equivalent (μg of QE/g).

**Table 2 molecules-27-01480-t002:** Total phenolics and flavonoids content of *Physalis angulata* fruit.

Extract	TPC (mg * GAE/g)	TFC (mg ** QE/g)
Petroleum ether	75.34 ± 4.6	40. 78 ± 4.3
Ethyl acetate	69.41 ±3.6	100.83 ± 4.2
Ethanol	106.54 ±3.5	130.48 ± 2.6

* Total phenolics content is expressed in terms of Gallic acid equivalent (μg of GA/g). ** Total flavonoids content is expressed in terms of quercetin equivalent (μg of QE/g).

**Table 3 molecules-27-01480-t003:** Chemical profiling of ethanolic extract (fruit) of *Physalis angulata* by using GC-MS analysis.

Retention Time (Min)	Molecular Formula	Molecular Mass (g/mol)	Peak Area (%)	Compound Name	Structure
37.743	C_12_H_24_O_2_	200.3178	0.469	Lauric acid	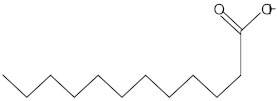
52.465	C_17_H_34_O_2_	270.4504	0.116	Methyl hexadecanoate	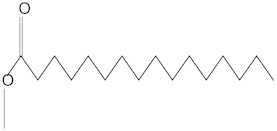
54.198	C_16_H_32_O_2_	256.4009	0.134	Palmitic acid	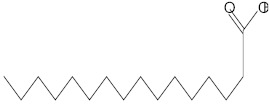
56.552	C_19_H_34_O_2_	294.4721	0.194	Linoleic acid, methyl ester	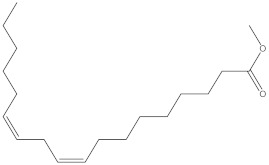
57.885	C_18_H_32_O_2_	280.444	1.774	Linoleic acid	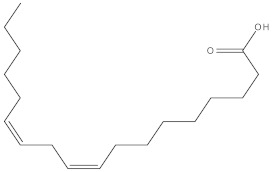
64.580	C_24_H_38_O_4_	390.5566	0.072	1,2-Benzenedicarboxylic acid, diisooctyl ester	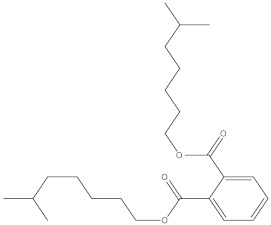
65.676	C_25_H_34_O_7_	446.5181	0.203	(22R)-6α,11β,21-Trihydroxy-16α,17α-propylmethylenedioxypregna-1,4-diene-3,20-dione	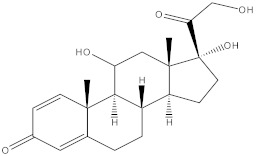
68.131	C_20_H_32_O	288.4702	0.215	5-(7a-Isopropenyl-4,5-dimethyl-octahydroinden-4-yl)-3-methyl-pent-2-enal	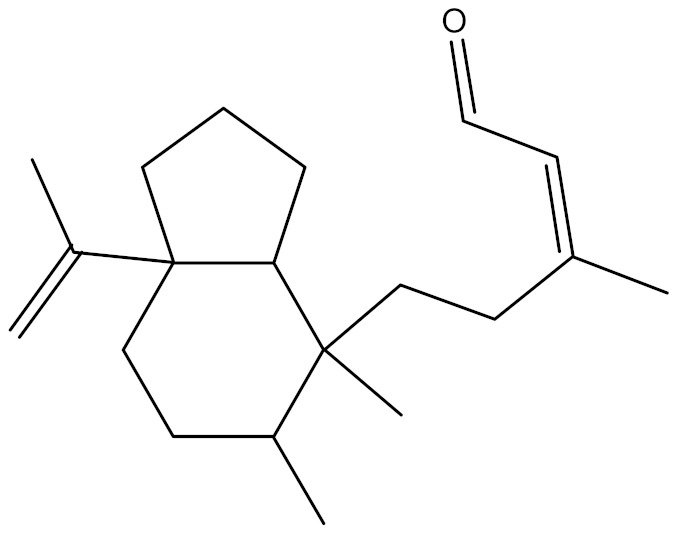
68.199	C_32_H_54_O_3_	486.8045	0.134	Acetic acid, 13-hydroxy-4,4,6a,6b,8a,11,11,14b-octamethyldocosahydropicen-3-yl ester	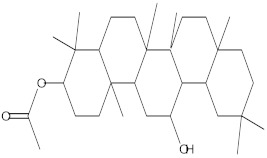
69.396	C_28_H_44_N_2_O_7_	520.6599	0.220	1-Pyrrolidinebutanoic acid, 2-[(1,1-dimethylethoxy)carbonyl]-α-nitro-, 2,6-bis(1,1-dimethylethyl)-4 -methoxyphenyl ester, [S-(R*,R*)]-	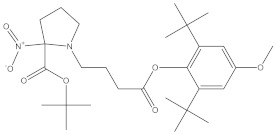
70.637	C_29_H_44_O_5_	472.6572	2.763	Cholest-5-ene-16,22-dione, 3β,26-dihydroxy-, 3-acetate, (20S,25R)-	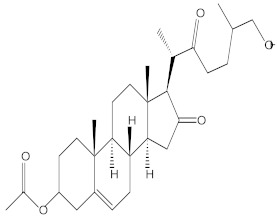
71.299	C_30_H_50_O_2_	442.7171	0.291	Ergost-5-en-3-ol, acetate, (3β,24R)-	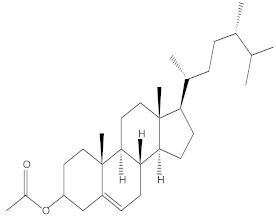
72.727	C_32_H_52_O_2_	468.7550	2.244	Lup-20(29)-en-3-ol, acetate, (3β)-	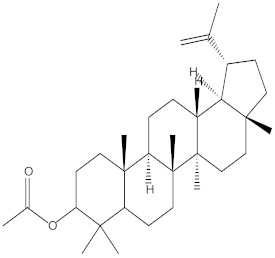
73.330	C_29_H_50_O_2_	430.7073	3.853	α-Tocopherol	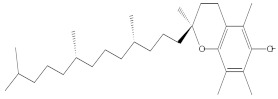

**Table 4 molecules-27-01480-t004:** Chemical profiling of ethanolic extract (leaves) of *Physalis angulata* by using GC-MS analysis.

Retention Time (Min)	Molecular Formula	Molecular Mass g/mol	Peak Area (%)	Compound Name	Structure
49.849	C18H36O	268.4780	3.51	Hexahydrofarnesyl acetone	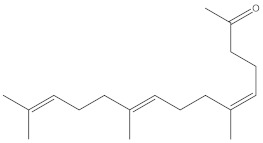
51.072	C20H40O	296.5315	2.055	3,7,11,15-Tetramethyl-2-hexadecen-1-ol	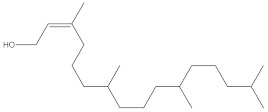
57.095	C20H40O	296.5315	10.192	Phytol	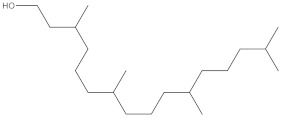
58.548	C18H34O2	282.4628	35.017	Oleic Acid	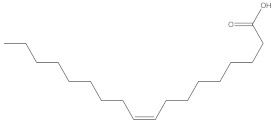
58.896	C18H36O2	284.4772	5.908	Octadecanoic acid	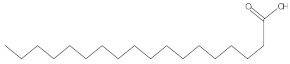
58.998	C23H39NO2	361.5627	0.747	3-[(1,5-Dimethyl-hexylamino)-methyl]-5,8a-dimethyl-3a,5,6,7,8,8a,9,9a-octahydro-3H-naphtho[2,3-b]furan-2-one	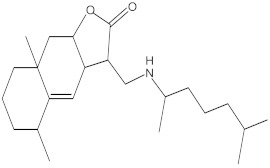
59.848	C18H32O2	280.4464	2.673	9,12-Octadecadienoic acid (Z,Z)-	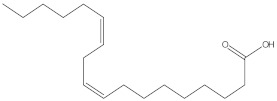
60.442	C14H24O2	224.3391	0.185	Linalyl isobutyrate	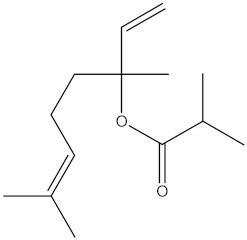
62.796	C21H38O4	354.5247	0.281	9,12-Octadecadienoic acid (Z,Z)-, 2,3-dihydroxypropyl ester	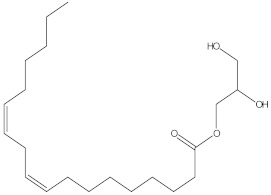
62.906	C21H40O3	340.5419	0.370	Oleic acid, 3-hydroxypropyl ester	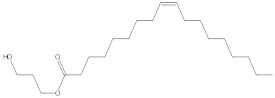
63.586	C22H40O2	336.5524	0.765	Butyl 9,12-octadecadienoate	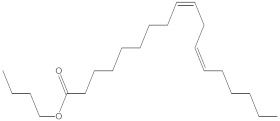
63.688	C21H40O4	356.5406	2.726	9-Octadecenoic acid (Z)-, 2-hydroxy-1-(hydroxymethyl)ethyl ester	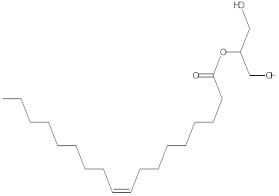
64.095	C21H40O3	340.5415	0.605	Glycidyl stearate	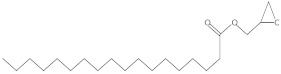
64.554	C24H38O4	390.5574	0.205	1,2-Benzenedicarboxylic acid, diisooctyl ester	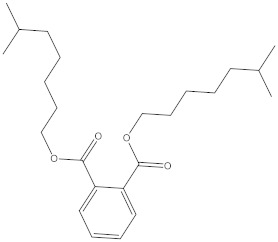
66.304	C42H88O5Si5	813.5739	0.607	5β-Cholestane-3α,7α,12α,24α,25-pentol TMS	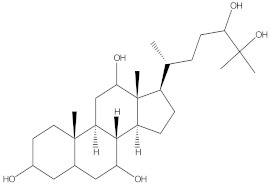
68.708	C30H50	410.7198	0.940	Squalene	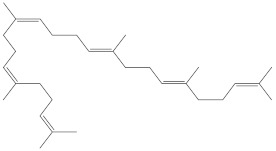
69.040	C40H66	546.9531	0.639	Lycopersene	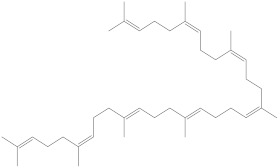
69.413	C28H44N2O7	520.6594	0.750	1-Pyrrolidinebutanoic acid, 2-[(1,1-dimethylethoxy)carbonyl]-α-nitro-, 2,6-bis(1,1-dimethylethyl)-4-methoxyphenyl ester, [S-(R*,R*)]-	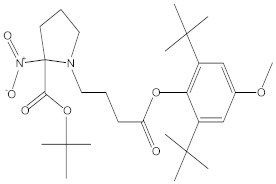
70.654	C30H44O2	436.6709	0.180	Anthiaergosatn-5,7,9,22-tetraen, 3-acetoxy-	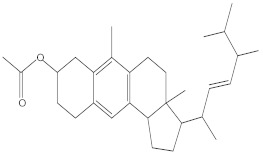
71.180	C29H48O	412.6919	0.325	Stigmasterol	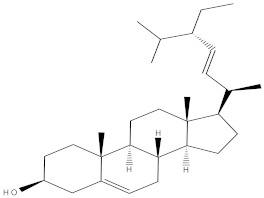
71.308	C30H50O2	442.7172	1.029	Ergost-5-en-3-ol, acetate, (3β,24R)-	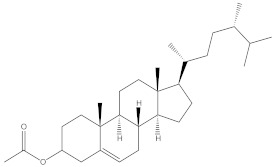
71.716	C31H50O2	454.7287	0.240	Stigmasta-5,22-dien-3-ol, acetate, (3β,22Z)-	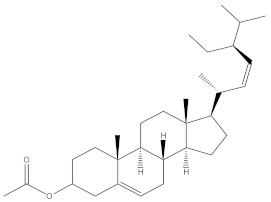
72.744	C29H48	396.6925	0.699	Stigmastan-3,5-diene	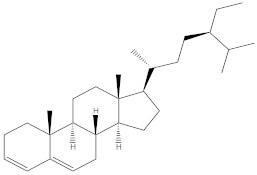
73.695	C29H50O2	430.7075	0.716	α-Tocopherol	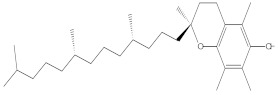

**Table 5 molecules-27-01480-t005:** Elemental analyses of *Physalis angulata* leaves and fruit by ICP-OES.

**Macroelements**			
**SI No**	**Elements**	***Physalis Angulata* Leaf (mg/kg)**	***Physalis Angulata* Fruit (mg/kg)**
1	Calcium	24,503	4143
2	Chromium	4.0	1.3
3	Potassium	35,725	32,856
4	Magnesium	4551	2823
5	Phosphorus	4642.5	5836.4
**Microelements**			
	**Elements**	***Physalis Angulata* Leaf (mg/kg)**	***Physalis Angulata* Fruit (mg/kg)**
1	Boron	30.0	20.1
2	Cobalt	˂0.1	˂0.1
3	Copper	27.7	9.8
4	Iron	3338.8	478.9
5	Manganese	48.6	15.7
6	Molybdenum	2.3	1.2
7	Sodium	1213	398.6
8	Nickel	4.1	2.5
9	Vanadium	1.5	1.2
10	Zinc	17.5	17.5
11	Aluminum	4713.8	716.2
**Other elements**			
1	Silver	0.3	0.5
2	Arsenic	˂0.1	˂0.1
3	Barium	76.1	10.0
4	Beryllium	˂0.1	˂0.1
5	Cadmium	0.4	0.4
6	Lead	1.4	˂0.1
7	Tin	˂0.1	˂0.1
8	Strontium	59.8	11.5

**Table 6 molecules-27-01480-t006:** Antibacterial activity of *Physalis angulata* leaves and fruit.

	*Physalis Angulata* Leaves		*Physalis Angulata* Fruit	
** *E. coli* **				
**Extract**	**MBC (mg)**	**MIC (mg)**	**MBC (mg)**	**MIC (mg)**
Petroleum ether	5	10	1.25	2.5
Ethyl acetoacetate	1.25	2.5	1.25	2.5
Ethanol	5	10	5	10
** *S. aureus* **				
Petroleum ether	5	10	5	10
Ethyl acetoacetate	1.25	2.5	5	10
Ethanol	2.5	5	5	10

MIC–Minimum inhibitory concentration (mg/mL), MBC–Minimum bactericidal concentration (mg/mL).

## Data Availability

Not applicable.

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
