# Peer review of "Chemical Composition Analysis, Cytotoxic, Antimicrobial and Antioxidant Activities of *Physalis angulata* L.: A Comparative Study of Leaves and Fruit"

_molecules, 2022, doi:10.3390/molecules27051480_

Round 1
Reviewer 1 Report
Dear Respected authors,
I am so pleased to scrutiny the content of this paper. I carefully checked the manuscript, and its organization, structure, and novelty of presented data were perfect. The discussed idea will provide further studies to deal with medicinal plants. The authors detailed the chemical profile of the studied plant and highlighted its biological activity. The literature requires such excellent papers to improve the quality of scientific publications—great congratulations on this publication. However, some comments should be addressed before publishing this manuscript.
- Please add the image of the Physalis angulate plant to the introduction section.
- Lines 30-38: Please update the references and consider relevant authorities for this section.
- Lines 57-60: please add the chemical structure of formerly identified compounds in this plant to the plant figure in comment 1.
- Lines 76-79: Please correctly mention the name of phenolics or alkaloids present in this plant. You can add an illustration to this section for including these secondary metabolites.
- Please provide a transparent flowchart for summarizing the M&M section. This will help academic readers to understand its procedure in detail.
- Lines81-88: Please compare the results of the DPPH assay with relevant references on this plant, if any. This helps the authors discuss this plant’s antioxidant effects in detail.
- Please unite figures 1-4 in one figure. All DPPH assay results should be in a united figure.
- Please increase the resolution of the figures present in figure 5. Using an optimized invert microscope, the respected authors can increase the resolution of given figures.
- Lines 133-137: Please provide the performed ANOVA analysis for discussed statistics.
- Please give a supplementary file for table 3 regarding the antibacterial assay in this study. If possible, please add the figure of plates where MIC values are taken from. This helps readers to see the qualitative results of your experimental assays.
- Please add GC-MS analysis chromatograms for all identified compounds in supplementary files.
- The most crucial part of this paper is table 5/6, where the respected authors discussed the identified compounds in the studied plant. For this section, to validate the results of antibacterial assays, please select some bacterial receptors/enzymes which might interact with these analyzed components, then try to conduct a protein-ligand docking analysis to unravel the binding mode of these compounds against bacterial receptors. This docking section will increase the quality of your paper and will provide extra data on the observed experimental antibacterial results. The respected authors can use autodock vina, molegro, or raccoon docking tools for this purpose. To show the docked compounds’ binding mode, it is better than the authors using PoseView online server or 3D protein imager. For the anticancer activity of the studied compounds, the authors can select some protein targets involved in the pathogenesis of cancer to conduct a second docking analysis. After doing docking analysis, please try to compare the docking energies together to know for which effects (antibacterial or anticancer) the docked compounds showed higher binding affinities.
- The discussion section should be improved. Please use the available references to expand the discussion and provide more details on the studied plant and its effects for academic readers.
I hope all the best for the authors and wish to see the published version of this paper soon.
Regards,
Rasouli. H
Author Response
Responses to reviewer comments
We would take this opportunity to thank the esteemed reviewer who has read the manuscript and provided us with their valuable comments and suggestions that have helped us improve the manuscript considerably. We hope that we have modified the manuscript as per the suggestions and recommendations of the learned reviewers.
- The “GC-Ms analysis” in the title of this article is suggested to be revised as “Chemical composition analysis”.
Response: As suggested by the learned reviewer, the title of this article is has been revised as “Chemical composition analysis, cytotoxic, antimicrobial and antioxidant activities of Physalis angulata L: A comparative study of leaves and fruits”.
- The full names of abbreviations (TPC and TFC) should appear in the first use.
Response: As suggested by the learned reviewer, full names of abbreviations of TPC and TFC have been incorporated in the revised manuscript.
- Abbreviations for plant names (P angulate) should be in italics.
Response: As recommended by the learned reviewer, the name of the plant (P angulate) is entered in italic font.
- Concentration units should be added in all the figure.
Response: As suggested by the learned reviewer, the Concentration units (µg/mL) have been added in the revised manuscript.
- Some captions are missing in figure. 1, please add.
Response: As recommended by the learned reviewer, the captions have been modified in the figure. 1.
- The blank control and positive control should be added in the cytotoxicity study to better explain the effect of the extract.
Response: As suggested by the learned reviewer, the blank control and positive control of all the three cell lines MCF-7, DLD-1 HeLa have been incorporated in the revised manuscript.
- Molecular mass should be specific to 4 decimal places in GC-MS analysis.
Response: As suggested by the learned reviewer, the molecular mass has been written to 4 decimal places in GC-MS analysis.
- It is suggested that the manuscript should first introduce the unified chemical composition analysis before introducing biological activity.
Response: As suggested by the learned reviewer, the manuscript has been rearrangement chemical composition analysis is introduced, followed by biological activity.
Reviewer 2 Report
This manuscript describes a comparative study of Physalis angulata leaf and fruit extracts, such as chemical composition analysis, anticancer, antimicrobial, antioxidant activity. To further investigations and isolation can be done for the leaf of Physalis angulate. However, it may be considered for publication after major revisions, the following comments should be solved before it is accepted.
- The “GC-Ms analysis” in the title of this article is suggested to be revised as “Chemical composition analysis”.
- The full names of abbreviations (TPC and TFC) should appear in the first use.
- Abbreviations for plant names (P angulate) should be in italics.
- Concentration units should be added in all the figure.
- Some captions are missing in figure. 1, please add.
- The blank control and positive control should be added in the cytotoxicity study to better explain the effect of the extract.
- Molecular mass should be specific to 4 decimal places in GC-MS analysis.
- It is suggested that the manuscript should first introduce the unified chemical composition analysis before introducing biological activity.
Author Response

(The authors gave the same response as above.)

Round 2
Reviewer 1 Report
Dear authors,
Thanks for submitting the revised version. I have no further comments on this paper. I hope to see your published manuscript soon.
Regards,
H. Rasouli
Reviewer 2 Report
Accept in present form